# Impact of the 2024 Korean medical workforce crisis on transfers in a pediatric emergency center: including comparative analyses with adults

Sung-Ha Kim[ID]©, Jin Hee Kim[ID]©, Jae-Hyun Kwon, Soo Hyun Park, Min-Jung Kim[ID], Young-Hoon Byun[ID], Ho-Young Song[ID], So-Hyun Paek[ID]*

Department of Emergency Medicine, Bundang CHA Medical Center, CHA University, Seongnam, South Korea

© These authors contributed equally to this work and share first authorship
* hyun21400@chamc.co.kr

## Abstract

### Background

In February 2024, the South Korean government announced a substantial increase in medical school admissions, leading to the sudden departure of most junior doctors. This unprecedented workforce shortage significantly impacted emergency departments. Pediatric care, known to require specialized readiness, may have been disproportionately affected.

### Methods

We conducted a retrospective study at an emergency institution with a specialized pediatric emergency center. The study period was divided into pre-policy (March 2023–January 2024) and post-policy (March 2024–January 2025) periods. All emergency visits were analyzed, and interhospital transfers were compared between periods. Interrupted time series (ITS) analysis, multivariable logistic regression and inverse probability of treatment weighting were performed to evaluate associations between the policy-driven workforce shortage and transfer occurrence. Subgroup analyses of transferred pediatric patients examined total length of stay, transfer reasons, and accompaniment by medical staff.

### Results

Among 97,916 ED visits (46,165 pediatric; 51,751 adult), there were 48 and 89 pediatric transfers in the pre- and post-policy periods, respectively, and 130 and 83 adult transfers in the same periods. Notably, pediatric transfers nearly doubled (an 85% increase) despite an approximately 42% decline in pediatric ED visit volume (29,211–16,954 visits). ITS analysis showed a continuing upward trend in pediatric transfers,

**Data availability statement:** All data underlying the findings described in this study are available from Zenodo at https://doi.org/10.5281/zenodo.17823408. Anonymized datasets and aggregated monthly transfer data are publicly available. The original non-anonymized clinical data cannot be shared publicly due to restrictions imposed by the Institutional Review Board of Bundang CHA Medical Center. Researchers who meet the criteria for access to confidential data may request access via the Bundang CHA Medical Center IRB (irb@chamc.co.kr).

**Funding:** The author(s) received no specific funding for this work.

**Competing interests:** The authors have declared that no competing interests exist.

although this trend did not reach statistical significance. In multivariable regression, only pediatric transfers were significantly associated with the post-policy period (OR 2.49, 95% CI 1.74–3.61). Transfer reasons shifted toward the unavailability of specialized care and physician accompaniment during transfers increased significantly.

## Conclusions

This study evaluated and identified the potential association between pediatric transfers and the government policy announcement, as well as the subsequent workforce departure, using statistical analyses and adjustment methods. Compared to adults, departure of the workforce following the policy announcement was more potentially associated with rises in pediatric transfers. Also, periodic changes in characteristics of pediatric interhospital transfers were observed. These findings reflect the vulnerability and complexity of pediatric emergency care and emphasize the necessity of ensuring medical readiness and specialized workforce capacity during times of healthcare disruption.

## Introduction

In 2024, the South Korean government attempted to increase the number of medical school admissions, attributing certain shortages of medical services to the limited supply of medical doctors. On February 6, government officials, including the President, announced the increase of admissions beginning in 2025, and then on April 11, the government emphasized that this figure represented the minimum increase necessary [1]. This policy corresponded to a 67% rise compared with the existing annual quota of 3,058 admissions.

Most medical students and junior doctors, including interns and residents, left their working environment in response to this unforeseen announcement. On February 19, approximately 9,000 residents at major hospitals collectively resigned and withdrew from clinical duties [2]. The presence of emergency medicine residents is known to enhance emergency department productivity and efficiency; thus, their sudden departure severely disrupted ED operations nationwide [3].

Interhospital transfers are well known to be multifactorial. Transfer decisions are influenced by multiple factors, encompassing not only clinical severity but also institutional capacity and patient-related social determinants [4]. Although no study has directly compared pediatric and adult transfer characteristics, pediatric transfers are widely recognized to require specialized readiness from medical professionals [5].

Hence, the aim of this study was to evaluate the impact of the sudden departure of the junior doctors' workforce following the government policy announcement, with a particular focus on transfers from our pediatric emergency center. In addition, a comparative analysis was performed with the adult group treated in the regional emergency medical center of the same institution.

## Methods

### Study design and data collection

This is a retrospective study was designed and conducted at single institution, a teaching hospital affiliated with CHA University, located in Gyeonggi Province (population 14.17 million; area 10,199 km²), South Korea. Within this institution, our pediatric emergency center is one of 12 pediatric emergency centers designated by the Ministry of Health and Welfare and cares for approximately 20,000 children under the age of 15 annually.

To assess the impact of the policy announcement and the subsequent workforce shortage, we divided the study period into two distinct phases: pre-policy and post-policy announcement. In February 2024, the South Korean government announced the policy, and shortly thereafter, most junior medical personnel, including interns and residents, left our institution around February 19, 2024. Therefore, we defined the two study periods as March 2023 to January 2024 (pre-policy announcement) and March 2024 to January 2025 (post-policy announcement), ensuring that the same months were compared to minimize seasonal and monthly variations in emergency department (ED) visits. February was excluded from the study period because we deemed it a transitional time. Each study period therefore comprised 11 months and data collection was confirmed to be complete through January 2025.

To explore periodic differences in the basic characteristics of the study population, univariate analyses were performed for all patients. Baseline characteristics included sex, age, transfer occurrence, initial severity or acuity using Korean Triage and Acuity Scale (KTAS) score, mode of arrival to the ED (Emergency Medical Services (EMS) vs. non-EMS), day type (weekday vs. weekend/holiday), and time of arrival (daytime 8 am–5 pm vs. night/weekend/holiday). The Korean Triage and Acuity Scale (KTAS) scores as follows: 1 (resuscitation), 2 (emergency), 3 (urgent), 4 (semi-urgent) and 5 (non-urgent). The primary analysis for the comparison used an interrupted time series (ITS) approach to evaluate level and trend changes in transfer rates before and after the workforce crisis. To address potential confounding by patient-level characteristics, multivariable logistic regression and inverse probability of treatment weighting (IPTW) analyses were performed as secondary sensitivity analyses. Multivariable logistic regression was conducted with transfer occurrence as the outcome variable, incorporating all covariates from the univariate analysis, the period split (pre- vs. post-policy announcement), and International Classification of Diseases (ICD) diagnostic groups. Fisher's exact tests were used to compare the system of diagnosis using ICD categories. For the systemic exploration using ICD-10 subgroup analysis, R codes, which represent non-specific symptoms or mild complaints rather than definitive diagnoses, were excluded to reduce diagnostic heterogeneity and enhance clinical interpretability. Because individual patients could have multiple concurrent ICD-10 diagnoses, all diagnoses recorded for each visit were mapped to their corresponding ICD-10 system categories. For each system category, a binary indicator was created. For comparative analyses of transferred pediatric patients between the two periods, we considered variables from the univariate analysis along with time from arrival to transfer decision, time from transfer decision to discharge, total emergency department length of stay, region of the receiving facility (nearby city, within city, intercity), reason for transfer (lack of intensive care unit (ICU) bed, need for emergency surgery, unavailability of specialist care, transfer for follow-up care, guardian request, lack of general ward bed, or lack of isolation room), mode of transfer (EMS vs. non-EMS), presence of accompanying medical staff during transfer, and the type of accompanying staff (specialist, resident, nurse, emergency medical technician (EMT), or none required).

### Statistical analysis

For the univariate analysis, continuous variables were compared using Welch's t-test. Categorical variables were analyzed using Pearson's chi-square test.

The interrupted time series (ITS) analysis was performed using segmented regression on monthly transfer rates (modeled as the proportion of transfers among total ED visits, using a quasibinomial generalized linear model), stratified by pediatric and adult groups. Separate ITS models were constructed for the pediatric and adult series. The models estimated both the immediate level change at the intervention and the difference in slope between the two periods. Serial

correlation in Pearson residuals was evaluated with the Durbin–Watson and Breusch–Godfrey tests. When evidence of first-order autocorrelation was detected, heteroskedasticity- and autocorrelation-consistent (HAC, Newey–West) standard errors were applied and generalized least squares (GLS) models with a first-order autoregressive [AR(1)] correlation structure were additionally fit on logit-transformed monthly transfer rates using inverse-variance weights (1/n_visits). Influence diagnostics (Cook's distance) were used to identify visually influential months, and pre-specified sensitivity analyses were performed excluding these months. All ITS results are presented separately for pediatric and adult models, expressed as odds ratios (ORs) with 95% confidence intervals (CIs).

We performed multivariable logistic regression to identify variables associated with transfer occurrence, including the period indicator (pre- vs. post-policy announcement). Separate logistic regression models were constructed for adult and pediatric populations. Odds ratios (ORs) with 95% confidence intervals (CIs) were calculated for each variable. A p-value of less than 0.05 was considered significant. To further assess whether the policy effect differed between adults and children, we fitted an additional pooled logistic regression model including an interaction term between periods and age groups. The interaction term tested whether the change in transfer likelihood after the policy implementation differed by age group. Fisher's exact tests were performed to compare systemic categories of diagnosis using ICD codes. To address multiple testing, the Benjamini–Hochberg (BH) procedure was applied.

To mitigate potential confounding by differences in patient populations pre- and post-policy announcement, we additionally performed IPTW. Propensity scores for being in the pre- or post-policy announcement period were estimated using multivariable logistic regression, including baseline covariates such as age; sex; arrival mode (EMS vs non-EMS); initial acuity (KTAS category); visit-timing indicators (weekday vs weekend/holiday and daytime vs nighttime/weekend/holiday); and ICD-10 system categories. Stabilized weights were applied to construct a weighted pseudo-population, and weighted logistic regression was then performed with transfer occurrence as the outcome and the period as the primary exposure.

To assess and compare differences in the transferred pediatric group, transferred children from both periods were studied. In these analyses of transferred pediatric patients, continuous variables were compared using Welch's t-test, and categorical variables were analyzed using Pearson's chi-square or Fisher's exact test, depending on distribution.

All statistical analyses were conducted using R version 4.3.2 (R Foundation for Statistical Computing, Vienna, Austria). Data preprocessing and management were performed with the tidyverse suite. For descriptive and comparative statistics, the stats package was used, with additional support from gtsummary and finalfit for regression analyses and table generation. Visualization of the study population, descriptive results, and regression outputs were carried out with DiagrammeR, ggplot2, and ggpubr. For ITS analysis, segmented regressions were fitted using the nlme and lmtest packages, with sandwich applied to obtain robust standard errors and forecast for autocorrelation checks. IPTW was implemented using the survey and MatchIt packages. Fisher's exact tests with BH adjustment for multiple comparisons were conducted using the stats and multtest packages. All tables, figures, and analyses were independently developed and validated by one of the lead investigators, who holds dual expertise in pediatric emergency medicine and statistical data science.

### Ethics statement

This study was approved by the Institutional Review Board of CHA Bundang Medical Center (IRB No. 2024-03-005), which waived the requirement for informed consent due to the retrospective design and use of de-identified data. Although the investigators initially accessed medical records that contained identifiable information, all data were de-identified prior to analysis and no identifying information was retained or accessible to the authors during or after data collection. All methods were performed in accordance with the relevant guidelines and regulations.

### Results

A total of 97,916 patients visited our ED during the study period. Among all patients, 46,165 were managed in the pediatric center and 51,751 in the adult division. To explore periodic differences across variables, the study population was divided according to the period split. In the pediatric division, 29,211 patients were included in the pre-policy announcement

period and 16,954 in the post-policy announcement period. A total of 36,657 adult patients were included in the pre-policy announcement period and 15,094 in the post-policy announcement period. Subsequently, all patients were categorized according to whether they were transferred to other facilities. Among pediatric patients, the number of transfers increased from 48 in the pre-policy period to 89 in the post-policy period, representing an 85% increase despite an approximately 42% decline in pediatric ED visit volume. Among adult patients, 130 and 83 transfers occurred, respectively. For comparative analyses of transferred children between the two periods, one pediatric case with a few missing searched variables in medical records was excluded, resulting in 136 pediatric patients in the final analysis (Fig 1).

In the univariable analysis of adult patients, the mean age was higher in the post-period population, and the proportion of male patients also increased. The proportion of patients with initial KTAS scores of 4 and 5 become lower in the post-period, indicating a higher proportion of patients with more severe and acute conditions. In addition, a larger proportion of patients visited the ED during weekdays and daytime hours in the post-period compared to the pre-period. In the pediatric group, both the absolute number and the proportion of transferred patients increased in the post-period. While the proportion of EMS arrivals was higher in the post-period, the absolute number was lower compared to the pre-period. In the post-period, a larger proportion of pediatric patients visited the ED at nighttime, on weekends, or holidays (Table 1).

In the ITS analysis, the relative probability of transfer seemed to have been increasing after the policy announcement in the pediatric group compared to that in the adult group (Fig 2). However, no statistically significant changes were detected in either the pediatric or adult groups after the policy announcement. Neither the immediate level change nor the slope change was statistically significant in either group (Table 2 and 3). Residual diagnostics showed no evidence of positive serial correlation by Durbin–Watson, but the Breusch–Godfrey test suggested AR(1) in the adult series. When re-estimated using HAC (Newey–West) standard errors and, separately, GLS models with AR(1) correlation structures, effect estimates were consistent in direction and magnitude, and inference was unchanged. In sensitivity analyses excluding the visually influential month (December 2024), both pediatric and adult models showed stable estimates without statistical significance. However, the pediatric trend remained upward while the adult trend appeared to decline slightly, supporting the robustness but indicating divergent directional trends (Table 3; Fig 2).

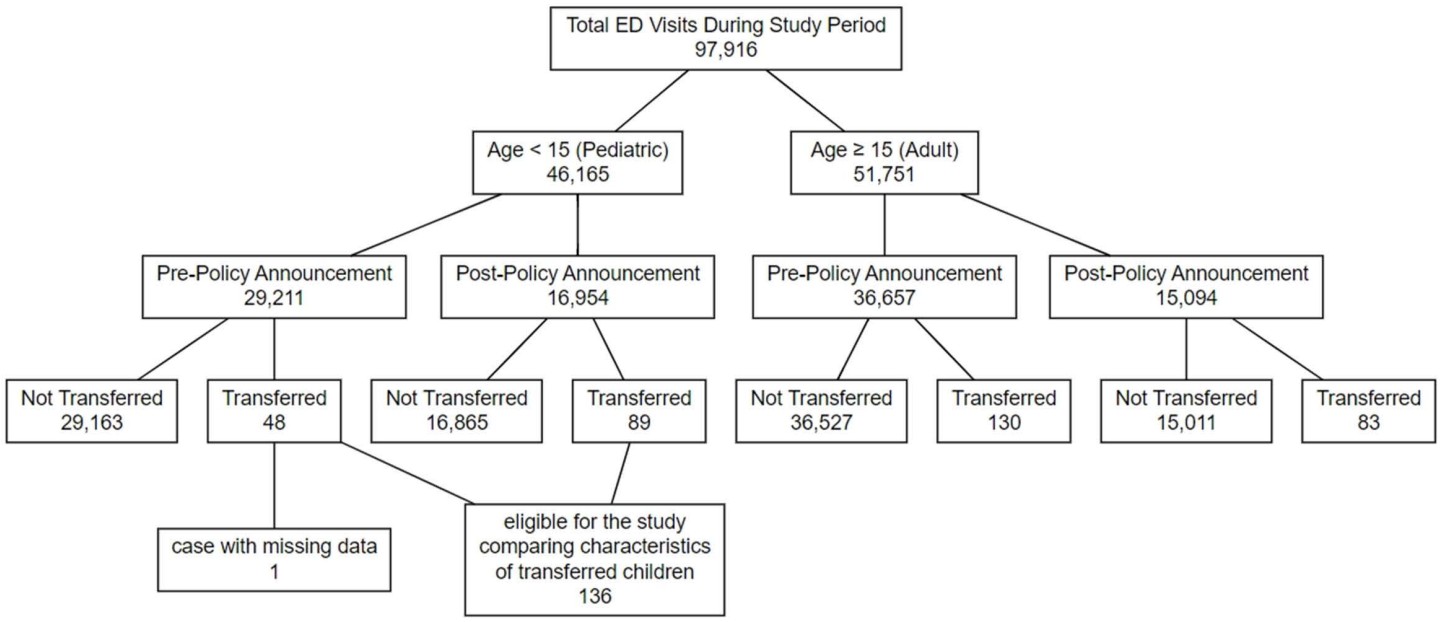

**Fig 1. Study population flow diagram.**

**Table 1. Baseline characteristics of the study population in the pre- and post-policy periods.**

| Variable | Adult | | Pediatric | |
|---|---|---|---|---|
| | Pre-Announcement N=36,657[1] | Post-Announcement N=15,094[1] | Pre-Announcement N=29,211[1] | Post-Announcement N=16,954[1] |
| **Age** | 53.2±20.3 | 58.2±19.6 | 3.9±3.4 | 3.8±3.7 |
| **Sex** | | | | |
| Male | 16,267 (44%) | 6,974 (46%) | 15,955 (55%) | 9,406 (55%) |
| Female | 20,390 (56%) | 8,120 (54%) | 13,256 (45%) | 7,548 (45%) |
| **Transfer** | | | | |
| Not transferred | 36,527 (100%) | 15,011 (99%) | 29,163 (100%) | 16,865 (99%) |
| Transferred | 130 (0.4%) | 83 (0.5%) | 48 (0.2%) | 89 (0.5%) |
| **Initial KTAS score** | | | | |
| KTAS 1 | 932 (2.5%) | 717 (4.8%) | 33 (0.1%) | 33 (0.2%) |
| KTAS 2 | 3,104 (8.5%) | 2,302 (15%) | 787 (2.7%) | 563 (3.3%) |
| KTAS 3 | 18,553 (51%) | 9,643 (64%) | 10,646 (36%) | 6,692 (39%) |
| KTAS 4 | 9,201 (25%) | 1,803 (12%) | 15,438 (53%) | 8,034 (47%) |
| KTAS 5 | 4,867 (13%) | 629 (4.2%) | 2,307 (7.9%) | 1,632 (9.6%) |
| **Arrival mode** | | | | |
| EMS | 9,478 (26%) | 4,188 (28%) | 1,605 (5.5%) | 1,324 (7.8%) |
| non-EMS | 27,179 (74%) | 10,906 (72%) | 27,606 (95%) | 15,630 (92%) |
| **Day type** | | | | |
| Weekday | 23,918 (65%) | 10,123 (67%) | 17,274 (59%) | 9,784 (58%) |
| Weekend/Holiday | 12,739 (35%) | 4,971 (33%) | 11,937 (41%) | 7,170 (42%) |
| **Arrival time** | | | | |
| Daytime | 11,134 (30%) | 4,896 (32%) | 5,377 (18%) | 2,895 (17%) |
| Nighttime/Weekend/Holiday | 25,523 (70%) | 10,198 (68%) | 23,834 (82%) | 14,059 (83%) |

[1]Mean±SD; n (%), [2]Welch Two Sample t-test; Pearson's Chi-squared test

KTAS, Korean Triage and Acuity Scale; EMS, Emergency Medical Services

In the multivariable logistic regression by transfer outcome, initial KTAS score (with KTAS 1 as the most severe reference category, such that lower acuity levels showed substantially reduced odds of transfer) and arrival mode via EMS were significantly associated with transfer occurrence in both the adult and pediatric groups. Female pediatric patients showed a higher transfer occurrence in the pediatric group. Notably, the periodic association was observed only in the pediatric group with statistical significance. The pediatric group from the post-period was more associated with transfer occurrence compared to those from the pre-period (OR 2.49, 95% CI 1.74–3.61, p<0.001) (Table 4). In the pooled model including both age groups, the interaction term between period and age group was statistically significant (OR = 2.44, 95% CI 1.56–3.85, p<0.001), indicating that the post-policy change in transfer rate was greater among pediatric patients compared with adults. With respect to diagnostic categories, ICD codes of infectious disease, endocrine/metabolic disease, and mental/behavioral disease were significantly associated with transfer outcomes in the adult group, whereas ICD codes of infectious disease, neoplasm/hematologic/immune disease, endocrine/metabolic disease, mental/behavioral disease, and digestive disease were associated with transfer outcomes in the pediatric group. Furthermore, no multicollinearity was observed across the variables (all VIFs<1.2).

However, when comparing diagnostic categories between the two periods in a more detailed Fisher's exact test, most raw p-values exceeded the conventional threshold for statistical significance in both the adult and pediatric groups, indicating the absence of a strong association. In the pediatric group, the respiratory category showed a potential association.

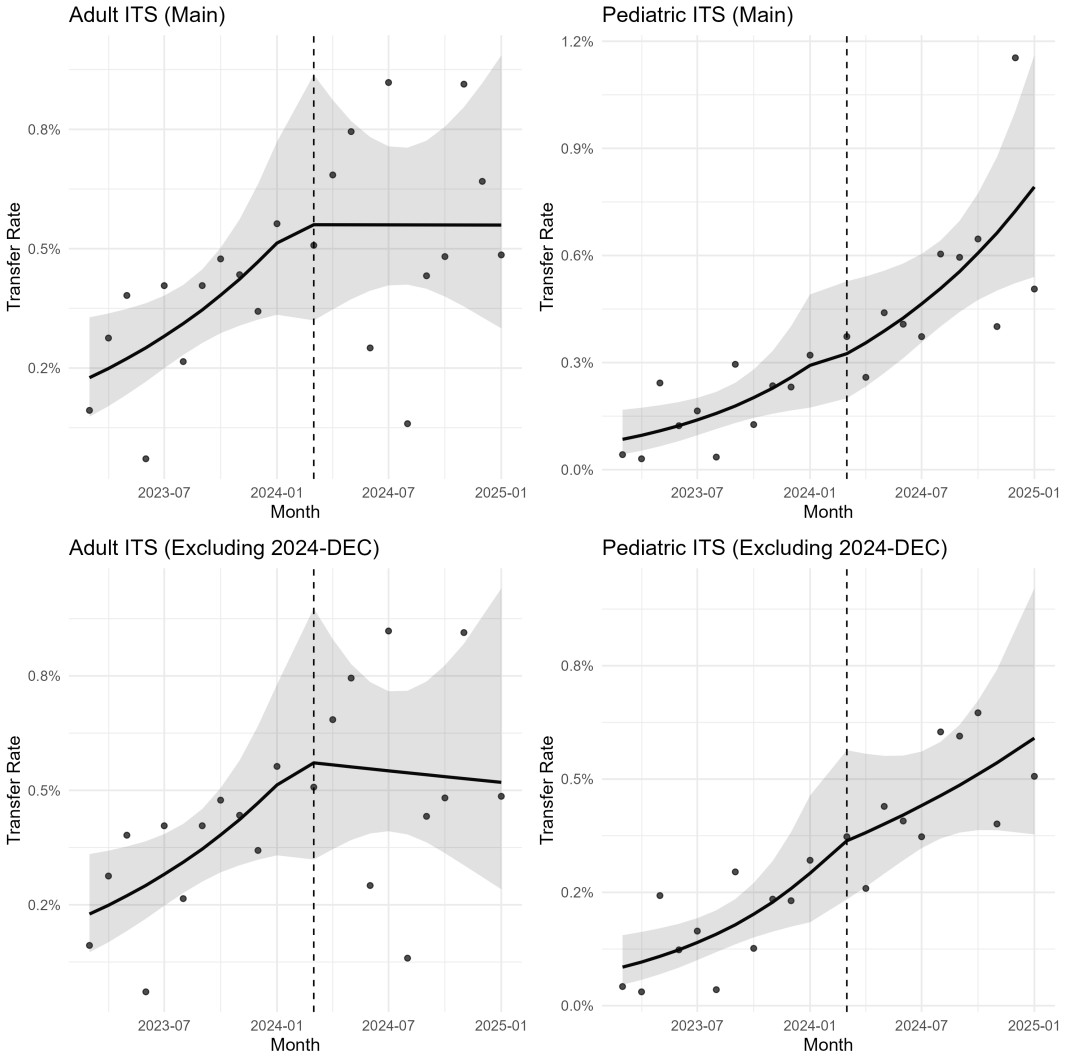

**Fig 2. Segmented regression plots from interrupted time series (ITS) analyses of emergency department transfers in adult and pediatric groups, including sensitivity analyses excluding the visually influential month (December 2024).** Monthly transfer rates are shown separately for both groups. The solid lines represent fitted segmented regression estimates for the pre-policy and post-policy periods, with shaded areas indicating 95% confidence intervals. The vertical dashed line marks the time of the policy announcement.

**Table 2. Interrupted time series (ITS) results — Main analysis.**

| Group | Change | OR | 95% CI | p value |
|-------|--------|-----|--------|---------|
| Pediatric | Level change | 1.017 | 0.476–2.173 | 0.966 |
| Pediatric | Slope change | 0.966 | 0.852–1.096 | 0.603 |
| Adult | Level change | 1.076 | 0.574–2.017 | 0.823 |
| Adult | Slope change | 0.923 | 0.831–1.025 | 0.149 |

**Table 3. Interrupted time series (ITS) results — Sensitivity analysis excluding the outlier-like-month (December 2024).**

| Group | Change | OR | 95% CI | p value |
|---|---|---|---|---|
| Pediatric | Level change | 1.186 | 0.597–2.356 | 0.632 |
| Pediatric | Slope change | 0.928 | 0.824–1.044 | 0.232 |
| Adult | Level change | 1.103 | 0.571–2.131 | 0.773 |
| Adult | Slope change | 0.916 | 0.816–1.027 | 0.150 |

CI, Confidence Interval; OR, Odds Ratio

**Table 4. Multivariable logistic regression of transfer outcomes in adult and pediatric groups.**

| Variable | Adult | | | | | Pediatric | | | | |
|---|---|---|---|---|---|---|---|---|---|---|
| | Unadjusted | | Adjusted | | | Unadjusted | | Adjusted | | |
| | OR | 95% CI | OR | 95% CI | p-value | OR | 95% CI | OR | 95% CI | p-value |
| **Sex (M vs F)** | 1.65 | 1.26, 2.17 | 1.30 | 0.99, 1.72 | 0.06 | 1.47 | 1.04, 2.11 | 1.50 | 1.05, 2.16 | **0.03** |
| **Age, years** | 1.01 | 1.01, 1.02 | 1.00 | 0.99, 1.01 | 0.8 | 0.99 | 0.94, 1.03 | 0.98 | 0.93, 1.02 | 0.3 |
| **Weekday** | 1.09 | 0.82, 1.46 | 1.09 | 0.79, 1.51 | 0.6 | 1.12 | 0.80, 1.59 | 0.84 | 0.56, 1.26 | 0.4 |
| Weekend/holiday | — | — | — | — | | — | — | — | — | |
| **Daytime** | 0.91 | 0.68, 1.22 | 0.81 | 0.58, 1.13 | 0.2 | 1.64 | 1.10, 2.37 | 1.33 | 0.83, 2.09 | 0.2 |
| Nighttime/weekend/holiday | — | — | — | — | | — | — | — | — | |
| **Initial KTAS score** | | | | | **<0.001** | | | | | **<0.001** |
| KTAS 1 | — | — | — | — | | — | — | — | — | |
| KTAS 2 | 0.52 | 0.34, 0.81 | 0.63 | 0.40, 1.02 | | 0.07 | 0.03, 0.15 | 0.26 | 0.10, 0.69 | |
| KTAS 3 | 0.19 | 0.13, 0.29 | 0.29 | 0.18, 0.47 | | 0.02 | 0.01, 0.03 | 0.06 | 0.02, 0.15 | |
| KTAS 4 | 0.06 | 0.03, 0.11 | 0.08 | 0.04, 0.16 | | 0.01 | 0.00, 0.02 | 0.03 | 0.01, 0.09 | |
| KTAS 5 | 0.05 | 0.02, 0.11 | 0.06 | 0.02, 0.16 | | 0.00 | 0.00, 0.01 | 0.01 | 0.00, 0.04 | |
| **Arrival by EMS** | 4.23 | 3.22, 5.58 | 2.79 | 2.05, 3.81 | **<0.001** | 4.92 | 3.29, 7.18 | 2.56 | 1.55, 4.09 | **<0.001** |
| Arrival by non-EMS | — | — | — | — | | — | — | — | — | |
| **Post-policy announcement period** | 1.55 | 1.18, 2.04 | 1.19 | 0.89, 1.58 | 0.2 | 3.21 | 2.27, 4.59 | 2.49 | 1.74, 3.61 | **<0.001** |

CI, Confidence Interval; OR, Odds Ratio; KTAS, Korean Triage and Acuity Scale; EMS, Emergency Medical Services.

After applying the BH adjustment to account for multiple comparisons, only the differences in J codes (respiratory diseases) remained statistically significant (BH adjusted p-value = 0.015) in the pediatric group (Tables 5 and 6).

In the supplementary IPTW-weighted logistic regression model, the post-policy period was significantly associated with an increased likelihood of interhospital transport (OR = 2.21, 95% CI = 1.55–3.16, p < 0.001) (S1 Table), after adjusting for baseline characteristics including age, sex, arrival mode, and diagnostic categories (S1 Figure). This suggests that the policy announcement and the subsequent shortage of the workforce may have contributed to a higher interhospital transport rate among pediatric patients, independent of case-mix differences.

In the comparative analysis of transferred children, overall length of stay in the emergency department, time to transfer decision, and time from decision to actual discharge did not show significant differences between the two periods. However, the reasons for transfer demonstrated statistically significant changes across periods. Furthermore, children in the post-period more frequently required accompaniment by medical personnel during transfers, and the type of accompanying medical staff differed significantly between the two periods, as expected (Table 7).

**Table 5. Fisher's exact test of systemic categories in adult group.**

| System by ICD codes | N | Pre & Not Transferred | Pre & Transferred | Post & Not Transferred | Post & Transferred | OR (Fisher) | CI Low | CI High | p (BH-adjusted) |
|---|---|---|---|---|---|---|---|---|---|
| Infectious | 3,267 | 2,207 | 5 | 1,055 | 0 | 0.000 | 0.000 | 2.287 | 0.657 |
| Neoplasm, Blood/ Immune | 4,751 | 2,861 | 13 | 1,873 | 4 | 0.470 | 0.111 | 1.525 | 0.657 |
| Endocrine/ metabolic | 2,865 | 1,564 | 2 | 1,295 | 4 | 2.415 | 0.345 | 26.741 | 0.697 |
| Mental/ behavior | 2,436 | 1,504 | 5 | 924 | 3 | 0.977 | 0.151 | 5.033 | 1.000 |
| Neurologic | 1,645 | 947 | 3 | 691 | 4 | 1.827 | 0.308 | 12.512 | 0.697 |
| Circulatory | 5,956 | 3,315 | 23 | 2,598 | 20 | 1.110 | 0.577 | 2.118 | 0.975 |
| Respiratory | 4,844 | 3,201 | 22 | 1,616 | 5 | 0.450 | 0.133 | 1.221 | 0.657 |
| Digestive | 7,574 | 4,606 | 22 | 2,931 | 15 | 1.071 | 0.516 | 2.165 | 0.975 |
| Genito-urinary | 5,660 | 3,683 | 10 | 1,959 | 8 | 1.504 | 0.515 | 4.239 | 0.697 |

**Table 6. Fisher's exact test of systemic categories in pediatric group.**

| System by ICD codes | N | Pre & Not Transferred | Pre & Transferred | Post & Not Transferred | Post & Transferred | OR (Fisher) | CI Low | CI High | p (BH-adjusted) |
|---|---|---|---|---|---|---|---|---|---|
| Infectious | 6,238 | 3,447 | 5 | 2,777 | 9 | 2.234 | 0.671 | 8.495 | 0.404 |
| Neoplasm, Blood/ Immune | 100 | 55 | 1 | 40 | 4 | 5.414 | 0.511 | 275.265 | 0.404 |
| Endocrine/ metabolic | 578 | 243 | 3 | 328 | 4 | 0.988 | 0.166 | 6.806 | 1.000 |
| Mental/ behavior | 72 | 35 | 1 | 34 | 2 | 2.039 | 0.102 | 124.782 | 1.000 |
| Neurologic | 172 | 84 | 3 | 79 | 6 | 2.118 | 0.435 | 13.528 | 0.587 |
| Circulatory | 139 | 62 | 2 | 71 | 4 | 1.740 | 0.240 | 19.844 | 0.883 |
| **Respiratory** | **8,303** | **5,035** | **12** | **3,233** | **23** | **2.985** | **1.423** | **6.590** | **0.015** |
| Digestive | 1,288 | 600 | 10 | 657 | 21 | 1.917 | 0.856 | 4.598 | 0.404 |
| Genito-urinary | 908 | 506 | 1 | 399 | 2 | 2.534 | 0.131 | 149.795 | 0.880 |

CI, Confidence Interval; OR, Odds Ratio; ICD, International Classification of Diseases; BH, Benjamini–Hochberg

## Discussion

Although transfers from the ED are multifactorial, we successfully evaluated and identified the potential association between pediatric transfers and the government policy announcement, as well as the subsequent workforce departure, using detailed statistical analyses and multiple adjustment methods. Unlike previous studies, we conducted a direct comparative analysis between adult and pediatric groups, revealing that pediatric patients may be more vulnerable to changes in transfer rates when confronted with medical policy shifts and related workforce availability.

Regarding visit volumes, overall ED visits declined substantially between the two study periods, from 65,868–32,048 visits (a 48.7% decrease). The reduction was even greater in the adult group, both in absolute numbers and in proportion. Previous studies have also evaluated declines in ED utilization during workforce crises. In 2016, Furnivall et al. conducted a nationwide analysis of resident workforce departure triggered by government-driven policy changes, which lasted only 1–2 days. That study reported that ED utilization decreased by approximately 7.1% across all patients, regardless of age group [6]. In South Korea, a 19-day strike by resident physicians in 2020 resulted in a workforce shortage in clinical practice. A study of six teaching hospital EDs in Daegu reported that, during the strike period, the total number of visits decreased, whereas the proportion of high-acuity patients (KTAS levels 1–2) increased [7]. However, unlike these studies, which focused only on the short-term effects of strikes, our study is distinguished by analyzing the characteristics

**Table 7. Comparison of Characteristics of Transferred Pediatric Patients Between Pre- and Post-Policy Announcement Periods.**

| Characteristic | Overall N = 136[1] | Pre-announcement N = 47[1] | Post-announcement N = 89[1] | p-value[2] |
|---|---|---|---|---|
| Age(year) | 3.7 ± 4.3 | 3.7 ± 4.2 | 3.7 ± 4.4 | 0.927 |
| Total stay in Emergency Department in minutes | 485.4 ± 1,827.3 | 300.1 ± 177.2 | 583.2 ± 2,253.5 | 0.242 |
| Minutes between arrival and transfer decision | 212.1 ± 231.3 | 237.0 ± 127.4 | 199.0 ± 270.3 | 0.268 |
| Minutes between transfer decision to discharge | 273.3 ± 1,819.2 | 63.1 ± 124.7 | 384.2 ± 2,243.4 | 0.182 |
| Sex | | | | >0.999 |
| _Male_ | 49 (36.0%) | 17 (36.2%) | 32 (36.0%) | |
| _Female_ | 87 (64.0%) | 30 (63.8%) | 57 (64.0%) | |
| Severity/acuity by initial KTAS score | | | | 0.907 |
| KTAS 1 | 12 (8.8%) | 4 (8.5%) | 8 (9.0%) | |
| KTAS 2 | 20 (14.7%) | 6 (12.8%) | 14 (15.7%) | |
| KTAS 3 | 59 (43.4%) | 19 (40.4%) | 40 (44.9%) | |
| KTAS 4 | 43 (31.6%) | 17 (36.2%) | 26 (29.2%) | |
| KTAS 5 | 2 (1.5%) | 1 (2.1%) | 1 (1.1%) | |
| Arrival mode | | | | 0.252 |
| _EMS_ | 34 (25.0%) | 15 (31.9%) | 19 (21.3%) | |
| _non-EMS_ | 102 (75.0%) | 32 (68.1%) | 70 (78.7%) | |
| Weekday | | | | 0.861 |
| _Weekday_ | 84 (61.8%) | 30 (63.8%) | 54 (60.7%) | |
| _Weekend/Holiday_ | 52 (38.2%) | 17 (36.2%) | 35 (39.3%) | |
| Daytime | | | | 0.097 |
| _Daytime (8am-5 pm)_ | 36 (26.5%) | 17 (36.2%) | 19 (21.3%) | |
| _Nighttime/Weekend/Holiday_ | 100 (73.5%) | 30 (63.8%) | 70 (78.7%) | |
| Transferred region | | | | 0.827 |
| _Nearby City (Seoul)_ | 82 (60.3%) | 30 (63.8%) | 52 (58.4%) | |
| _Inner-city (Gyeonggi)_ | 51 (37.5%) | 16 (34.0%) | 35 (39.3%) | |
| _Inter-city_ | 3 (2.2%) | 1 (2.1%) | 2 (2.2%) | |
| Transfer reason | | | | **<0.001** |
| _No ICU bed_ | 35 (25.7%) | 8 (17.0%) | 27 (30.3%) | |
| _Need for emergency surgery_ | 16 (11.8%) | 10 (21.3%) | 6 (6.7%) | |
| _Definitive treatment unavailable (specialist absent)_ | 57 (41.9%) | 12 (25.5%) | 45 (50.6%) | |
| _Transfer to follow-up hospital_ | 9 (6.6%) | 7 (14.9%) | 2 (2.2%) | |
| _Requested by guardian_ | 5 (3.7%) | 2 (4.3%) | 3 (3.4%) | |
| _No general ward bed_ | 3 (2.2%) | 2 (4.3%) | 1 (1.1%) | |
| _No isolation room_ | 11 (8.1%) | 6 (12.8%) | 5 (5.6%) | |
| Transfer mode | | | | 0.960 |
| _EMS_ | 114 (83.8%) | 40 (85.1%) | 74 (83.1%) | |
| _non-EMS_ | 22 (16.2%) | 7 (14.9%) | 15 (16.9%) | |
| Medical staff company | 86 (63.2%) | 22 (46.8%) | 64 (71.9%) | **0.005** |
| Type of accompanied medical staff | | | | **<0.001** |
| _Specialist_ | 32 (23.5%) | 3 (6.4%) | 29 (32.6%) | |
| _Resident_ | 17 (12.5%) | 17 (36.2%) | 0 (0.0%) | |
| _Nurse_ | 10 (7.4%) | 0 (0.0%) | 10 (11.2%) | |
| _EMT_ | 54 (39.7%) | 20 (42.6%) | 34 (38.2%) | |
| _Not necessary_ | 23 (16.9%) | 7 (14.9%) | 16 (18.0%) | |

[1]Mean ± SD; n (%)

[2]Welch Two Sample t-test; Fisher's exact test; Pearson's Chi-squared test

KTAS, Korean Triage and Acuity Scale; ICU, Intensive Care Unit; EMS, Emergency Medical Services; EMT, Emergency Medical Technician

of patients who utilized the ED during a more prolonged period of medical workforce shortage, with a direct comparison between adult and pediatric groups. A recent study from a pediatric ED in South Korea also reported that, following the 2024 government announcement of medical school quota expansion, patient visits decreased by approximately 50%, while the proportion of high-acuity patients significantly increased, replacing low-acuity cases [8]. According to our results, although the absolute numbers of patients across KTAS categories declined, their proportions changed significantly, indicating that our ED became more responsible for higher-severity patients after the policy announcement and workforce departure.

These findings regarding visit volumes and patient severity are consistent with our results and may also help explain the observed differences in EMS usage and mean ages of adult patients between the two periods. An increase in the number of initially severe patients and a corresponding increase in EMS usage for ED arrival seem logical. In 2023, Peters et al. similarly reported that patients arriving via EMS in the United States showed significantly higher severity [9]. Together, our findings and those of previous studies suggest a possible chain-reaction–like association among increased EMS arrivals, greater initial severity, and workforce crises. The significant increase in the mean age of adult patients in the post-policy period may be explained by the relative rise in the proportion of cases with higher severity or acuity. Previous studies have demonstrated that age is strongly associated with severity and outcomes in adult ED patients [10]. In Korea, Kim et al. reported that older adults presented with higher acuity, admission, and mortality rates compared with younger patients, using the KTAS [11]. As overall ED visits declined and proportion of more severe and acute patients increased in our study, we believe that the epidemiology has shifted toward the patients who are more likely to be older

In our study, visit timing also revealed contrasting patterns between adult and pediatric groups. After the announcement and workforce crisis, adults were more often presented during daytime/weekday hours, whereas children more frequently arrived at night and on weekends, with statistical significance. To our knowledge, no previous studies of medical workforce crises have specifically examined changes in ED visit timing, such as daytime versus nighttime or weekday versus weekend/holiday arrivals. Prior reports, including one from the UK, mainly focused on overall ED utilization, patient acuity, and throughput indicators during physician strikes, without addressing temporal visit patterns [12]. In this context, our findings provide novel evidence that workforce departure may influence not only the volume and severity of ED visits, but also the timing of patient presentations, with contrasting patterns observed between adults and children. This may be due to lower overall accessibility in pediatric care compared to adult care in South Korea [13]. Further studies would be required and would be beneficial for future workforce distribution in the ED.

When it comes to changes in transfer, there was an increase in the number of both adult and pediatric transfers in the post-period. Overall, ITS showed an increase in slope and level in the pediatric group compared to the adult group. However, in the ITS evaluation, slope and level changes of transfer rate were not statistically significant throughout the study period. While point estimates suggested a steeper slope toward the end of the post-period, the 11-month study window may have been insufficient to detect modest slope changes. Therefore, the ITS findings should be interpreted with caution, as they may reflect emerging trends that could become more apparent over a longer observation period. These patterns may be explained by several plausible mechanisms. During the workforce shortage, emergency physicians may have adopted a more risk-averse decision-making approach, leading to a lower threshold for transferring pediatric patients. In addition, limited pediatric readiness in general emergency departments, including reduced availability of specialized personnel and resources, may have further contributed to the increased reliance on interhospital transfer. While these interpretations remain speculative, they are consistent with the observed shifts in transfer patterns. These findings are also consistent with the broader landscape following the 2024 crisis. A nationwide ITS study using insurance data similarly documented a sharp and sustained decline in ED visits after the junior physicians' walkout, with pediatric ED bed occupancy not returning to pre-crisis levels even during a subsequent COVID-19 resurgence [14]. Although that study focused on ED utilization rather than transfer outcomes, the convergent pattern of sustained disruption in pediatric emergency care supports the findings of our study.

Consistent with the ITS trend, multivariable logistic regression and IPTW analyses showed an association between transfer outcomes and the periodic difference in the pediatric group. Other severity-related variables, such as the initial KTAS score group and EMS arrival, were associated with transfer outcomes in both study groups. Notably, the post-policy increase in pediatric transfers remained significant even after adjustment for demographic and clinical covariates in the multivariable model. This suggests that the observed rise may not be solely attributable to shifts in patient characteristics but rather may reflect structural or policy-related factors influencing referral behavior. The apparent discrepancy between ITS and logistic regression or IPTW findings warrants explicit methodological explanation. ITS operates at the population level, modeling monthly aggregated transfer rates and explicitly accounting for temporal autocorrelation and pre-existing secular trends; therefore, the net policy effect captured by ITS reflects only the marginal change above background temporal dynamics. Moreover, with only 11 monthly data points per group, the statistical power of ITS is inherently limited for detecting modest changes. In contrast, multivariable logistic regression and IPTW operate at the patient level, comparing individual transfer probabilities while adjusting for case-mix confounders such as acuity, arrival mode, and diagnostic category. This patient-level adjustment directly isolates the association between the period indicator and transfer likelihood, independent of the overall reduction in ED visit volume. Furthermore, the substantial visit decline introduces a denominator effect that the ITS model partially absorbs but that does not attenuate the patient-level regression estimates. Together, the directional consistency between ITS trends and the statistical significance of patient-level analyses may strengthen the overall interpretation that the post-policy period was associated with an increase in pediatric transfer risk.

Our Fisher's exact test with BH adjustment revealed that only the respiratory system was associated with periodic change and transfer outcome in the pediatric group. Given that the change in transfer after the workforce crisis was evident only in the pediatric group, we sought to statistically verify it further and explore the characteristics of transferred children.

Interhospital transfers are well acknowledged as multifactorial [15]. Pediatric interhospital transfers not only reflect multifactorial decision-making but also tend to involve patients with higher severity and resource requirements. For example, Odetola et al. reported that pediatric patients transferred from level II to level I pediatric ICUs had higher mortality and greater utilization of advanced therapies, underscoring the complexity of transferred children [16]. To identify the effect of periodic change more accurately, we conducted ITS analysis and logistic regression using multiple variables, including various diagnostic systems defined by ICD codes. In addition, we verified the significant periodic association through the IPTW as a supplementary method in the pediatric group.

Differences in interfacility transfers between pediatric and adult groups have been studied in a few previous reports [17]. The disproportionate increase in transfers among pediatric patients may reflect the vulnerability of pediatric care in South Korea, where specialized pediatric resources are relatively limited compared to adult care [12]. Moreover, Newgard et al. demonstrated that higher levels of pediatric readiness in emergency departments were associated with significantly lower short- and long-term mortality, underscoring that pediatric care systems are more sensitive to variations in resource availability compared to adult care [18]. During a workforce shortage, emergency physicians may have adopted a more risk-averse approach, leading to a higher tendency to transfer pediatric patients. Furthermore, Lee et al. reported that children and adolescents with life-limiting or complex conditions required higher medical expenditures and intensive care utilization, highlighting the inherently more complex diagnostic and management processes in pediatric populations [19]. These findings suggest that, during periods of medical workforce departure, the limited readiness of general EDs to provide specialized pediatric care, combined with the higher complexity of pediatric diagnosis and management, could have contributed to the greater increase in transfers among children relative to adults.

Regarding the association we found between diagnostic system groups, periodic change, and transfer outcome, we suggest that interpreting our Fisher's exact test with BH adjustment requires caution. It is well established that most severe pediatric patients are diagnosed with respiratory diseases [20]. Although we applied the IPTW method to minimize the effect of volume differences across diagnostic groups, our study also showed a relatively large number of respiratory

diagnoses, which may have led to the statistically significant association. Further studies with larger and more balanced numbers across diagnostic categories will be required to better understand the association between diagnosis, chief complaint, and transfer outcome.

Specific characteristics of transferred children were also explored. Although total stay and time to the decision and discharge did not change with statistical significance, the median ED stay increased markedly from approximately 300 minutes in the pre-policy period to 583 minutes in the post-policy period. A more detailed analysis revealed that the time to transfer decisions was shorter in the post-policy period, whereas the time from decision to actual discharge was substantially prolonged. Previous research on the 2020 resident strike, which represented short-term workforce disruption, reported a significant decrease in emergency department length of stay compared to control periods, likely attributable to reduced unnecessary testing and more rapid disposition decisions [7]. Similarly, in our study, the shorter time to transfer decisions may reflect the direct involvement of pediatric emergency medicine specialists, who were able to make quicker decisions in the absence of trainees. Nevertheless, several pragmatic factors contributed to the prolonged interval from transfer decision to actual discharge. During the workforce shortage, most pediatric emergency centers were staffed by only one pediatric emergency physician, who had to provide both direct patient care and arrange transfer acceptance. And this burden applied to both sending and receiving medical staff, leading to failure of prompt confirmation of acceptance. Furthermore, critically ill pediatric patients often require medical staff accompaniment [21]. However, with the absence of interns, residents, and fellows, pediatric emergency physicians themselves had to accompany patients, occasionally waiting until shift changes, thereby prolonging the transfer process.

The qualitative reasons for transferred pediatric patients also shifted between the two study periods. While "definitive treatment unavailable" remained the most frequent reason for transfer, its proportion increased significantly from 25.5% in the pre-policy period to 50.6% in the post-policy period. This finding indicates that transfers were increasingly driven not by bed shortages or the need for emergency surgery, but rather by the lack of specialized personnel during the workforce crisis. Although transfers can be beneficial when clinically necessary, prolonged workforce shortages may also lead to transfers driven by physician burnout and the unavailability of specialists [22]. Such potentially avoidable transfers should not be overlooked in applying new medical policies or facing impeding workforce shortage in healthcare system, as they expose patients to risks inherent in the transfer process, impose additional costs, and may adversely affect outcomes.

Finally, notable yet predictable changes were observed in the accompaniment of medical personnel during transfers. In the pre-policy period, only 6.4% of pediatric transfers were accompanied by a specialist, whereas this proportion increased to 32.6% in the post-policy period. This shift directly reflects the absence of interns, residents, and fellows, necessitating that pediatric emergency physicians accompany patients themselves. However, specialist accompaniment during transfers places a substantial burden on workforce availability within the emergency department and may reduce the capacity to care for other pediatric patients remaining in the ED. This finding highlights an important systemic implication, that prolonged workforce shortages not only affect transfer patterns but also strain the in-hospital availability of specialized pediatric emergency care.

These findings collectively carry important policy implications. Abrupt and large-scale workforce policy changes, implemented without adequate transition planning or safeguards, risk creating disproportionate disruptions in vulnerable subgroups such as pediatric patients. In particular, any reform that substantially alters the composition of clinical training or practice pipelines should be accompanied by clear protocols to maintain pediatric emergency readiness, including minimum staffing thresholds and coordinated transfer support systems. More broadly, healthcare systems should designate pediatric emergency capacity as a protected resource during periods of workforce shortage, given the demonstrated sensitivity of pediatric emergency outcomes to variations in specialist and trainee availability [18].

This study has several limitations. Although our institution is a designated pediatric emergency center with a high patient volume in South Korea, this study was conducted at a single center, which may limit generalizability. In addition, the workforce crisis examined in this study was triggered by a unique nationwide policy change in South Korea, which

may limit the applicability of our findings to other healthcare systems or to workforce shortages arising from different contexts. Furthermore, unmeasured external factors, such as changes in regional referral patterns, hospital resource availability, or public healthcare-seeking behavior during the workforce crisis, may have influenced patient volumes and transfer outcomes. Second, although the ITS model accounts for temporal trends and variability, the substantial reduction in ED visits during the post-policy period may have reduced statistical power, potentially limiting the ability to detect modest effects. Furthermore, the retrospective design inevitably carries the risk of information bias. Time-related variables were defined using transfer consent timestamps from hospital records rather than internationally standardized criteria, and ICD classifications applied to multi-diagnosis visits may have introduced category overlap. Finally, compared to previous studies which mainly focused on clinical outcomes such as mortality [23], we were unable to evaluate post-transfer outcomes and thus could not determine whether transfers ultimately improved patient prognosis. Despite these limitations, our study provides meaningful insights into how systemic disruptions may differentially affect vulnerable groups, especially in children.

In conclusion, our study demonstrated that the sudden departure of junior doctors' workforce following a government-driven policy change was associated with an increase in interhospital transfers among pediatric patients compared to adults. Although overall ED utilization declined in both groups, pediatric patients experienced a significant rise in transfer rates, increasing from 48 to 89 transfers (an 85% increase) despite an approximately 42% decline in pediatric ED visit volume, reflecting the structural vulnerability and clinical complexity of pediatric emergency care. Furthermore, specific changes in characteristics of pediatric interhospital transfers were observed throughout the study periods. These findings indicate the importance of ensuring sufficient pediatric readiness and specialized workforce capacity, particularly during times of healthcare disruption.

## Supporting information

**S1 Table. Inverse probability of treatment weighting (IPTW)-adjusted logistic regression results for transfer outcomes in pediatric group.**
(DOCX)

**S1 Fig. Love plot of covariate balance before and after IPTW adjustment in the pediatric group.** The plot displays standardized mean differences (SMDs) for baseline covariates comparing pre- and post-policy announcement periods. Gray dots indicate covariate balance before weighting, and blue dots indicate balance after IPTW adjustment. The vertical dashed line at an absolute SMD of 0.1 represents the conventional threshold for acceptable covariate balance. ICD, International Classification of Diseases; KTAS, Korean Triage and Acuity Scale.
(TIFF)

## Author contributions

**Conceptualization:** Sung-Ha Kim, Jin Hee Kim, So-Hyun Paek.

**Data curation:** Sung-Ha Kim, Jin Hee Kim, Jae-Hyun Kwon, Soo Hyun Park, Min-Jung Kim, Young-Hoon Byun, Ho-Young Song.

**Formal analysis:** Sung-Ha Kim, Jin Hee Kim, So-Hyun Paek.

**Investigation:** Sung-Ha Kim, Jin Hee Kim, So-Hyun Paek.

**Methodology:** Sung-Ha Kim, Jin Hee Kim, Jae-Hyun Kwon, Soo Hyun Park, Young-Hoon Byun, Ho-Young Song, So-Hyun Paek.

**Project administration:** Sung-Ha Kim, Jin Hee Kim, So-Hyun Paek.

**Resources:** Sung-Ha Kim, So-Hyun Paek.

**Software:** Sung-Ha Kim.

**Supervision:** Sung-Ha Kim, Jin Hee Kim, So-Hyun Paek.

**Validation:** Sung-Ha Kim, Jin Hee Kim.

**Visualization:** Sung-Ha Kim.

**Writing – original draft:** Sung-Ha Kim, Jin Hee Kim.

**Writing – review & editing:** Sung-Ha Kim, Jin Hee Kim, Min-Jung Kim, So-Hyun Paek.

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
