## [Decision Letter · Decision Letter 0]

4 Nov 2025

PONE-D-25-47181Impact of the 2024 Korean medical workforce crisis on transfers in a pediatric emergency center: including comparative analyses with adultsPLOS ONE

Dear Dr. Paek,

Thank you for submitting your manuscript to PLOS ONE. After careful consideration, we feel that it has merit but does not fully meet PLOS ONE’s publication criteria as it currently stands. Therefore, we invite you to submit a revised version of the manuscript that addresses the points raised during the review process.

Thank you for submitting your manuscript to PLOS One. After a careful evaluation of your submission and consideration of the reviewers’ feedback, we invite you to revise your manuscript by addressing the issues and suggestions raised.Please provide a detailed, point-by-point rebuttal outlining your responses to each reviewer comment, along with a tracked version of the revised manuscript highlighting all changes made.

We look forward to receiving your revised submission for further consideration.

We look forward to receiving your revised manuscript.

Kind regards,

Sonu Bhaskar, MD PhD

Academic Editor

PLOS ONE

Journal Requirements:

2. In the online submission form, you indicated that [The dataset analyzed in this study is not publicly available due to the policy of the institutional review board of Bundang CHA Medical Center. However, datasets are available from the corresponding author upon reasonable request.].

Reviewers' comments:

Reviewer's Responses to Questions

**Comments to the Author**

1. Is the manuscript technically sound, and do the data support the conclusions?

Reviewer #1: Partly

Reviewer #2: Partly

2. Has the statistical analysis been performed appropriately and rigorously? 

Reviewer #1: Yes

Reviewer #2: Yes

3. Have the authors made all data underlying the findings in their manuscript fully available?

Reviewer #1: No

Reviewer #2: Yes

4. Is the manuscript presented in an intelligible fashion and written in standard English?

Reviewer #1: No

Reviewer #2: Yes

5. Review Comments to the Author

Reviewer #1: The authors present 3 inter-related analyses of hospital transfers before and after a key policy change which led to an acute workforce shortage among South Korean physicians in February 2024. Interrupted time series did not find any significant change in pediatric transfers after the policy change (though there did appear to be a pre-existing upward trend); in contrast, multivariable logistic regression and IPTW did find that the post-policy time period seemed to be associated with an increase in transfer rate (allbeit no longer accounting for the pre-existing upward trend).

I think the work is overall an interesting opportunity to analyze the real world effects of a discrete policy change. It is both well suited to time series analysis and I think reasonable/appropriate to include at least one other analysis to adjust for potential changes in patient traits over time - though I think conducting both IPTW and multivariate regression is a bit duplicative as both adjust for this issue. The exploration of patterns of systemic disease categories pre and post policy change becomes quite speculative/exploratory, though at least p values were aggressively adjusted for (BH method given risk of false discovery).

I do think a number of areas could benefit from extra clarification/cleanup:

1) On selection of 3 different analytical methods: This work certainly seems best suited to interrupted time series analysis (which was one of the methods undertaken). I am a bit curious about the specific reasons for using both of the other methods. To me it would seem that multivariable logistic regression and IPTW effectively seek to accomplish similar goals (e.g., assessing likelihood of transfer by pre- or post-policy period while adjusting for potential confounders among patient-level characteristics).

-I think interrupted time series makes good sense as the primary analysis method

-I would next typically pick either multivariate logistic regression or IPTW (rather than both) as the second analysis - and frame it more clearly as an attempt to address/minimize potential confounding by differences in patient populations.

-If the authors really want to show the effect of potential confounders, then the multivariate regression approach might make the most sense as it more explicitly shows these effects; if the authors are most interested in simply looking at the overall trend in transfers (which seems to be their stated goal), then IPTW alone might make sense.

-If the authors really wanted to present results from both multivariate logistic regression and IPTW, I would still pick one or the other for the main manuscript and present the other as an alternative approach (perhaps in Supplemental materials) more as a sensitivity analysis showing that both come up with similar results. This would avoid making it seem as though the two are truly fully separate analyses when I don't see them as such (I suspect the IPTW weighting model actually closely resembles the logistic regression model).

-On that note, I was not clear what variables were utilized for IPTW - it would be worth specifying

2) Appreciate the specific reference to assessment for auto-correlation and use of robust standard errors (lines 133-135), though the authors do not actually mention whether there was evidence of auto-correlation. If there were evidence of autocorrelation, there are usually additional modeling steps needed to account for that.

3) Line 137: Would soften/avoid language such as "To verify causal inference" as no adjustment can ever fully verify causal inference. Might suggest alternative language such as "To mitigate or address potential confounding by differences in patient populations pre- and post-policy announcement..."

4) Table 2: The presentation of Table 2 makes me begin to wonder how age grouping was handled (Adult vs Pediatric) - were two separate models constructed for each age group? Or was age group considered as a variable?

-If part of the point of the article were that adult and pediatric trends differed, one would usually consider something like an interaction term to statistically test whether adult/pediatric trends differed. This would usually provide justification for producing separate models.

-Regardless of how age group were handled, many statistically inclined readers might like to see separate columns displaying the univariate (unadjusted) and multivariate (adjusted) ORs as this would help the reader understand the effect of adjustment for suspected confounders.

5) Table 3A: I would suggest only showing the adjusted p values as these are more exploratory/subgroup based analyses.

6) Table 4: similar to above, I am curious if two separate models (adult and pediatric) were set a priori or if there were specific interest in demonstrating their trends differed (in which case again an interaction test might help assess whether these are statistically significantly different from one another).

7) Figure 2: There is a single month near what appears to be "2025-12" with a near 1.2% transfer rate. As this point at least visually looks as though it could serve as an influential outlier, consider sensitivity analysis of trend with exclusion of this point.

8) In discussion of results, some of the above suggestions regarding methods might also apply:

-As expected, it seems IPTW and multivariate logistic regression came to the same conclusion that rate of pediatric transfers increased post policy change. This would seem to support that these are not really fully independent analyses (as in comment 1 above), but perhaps ought to have only one presented as a main analysis and the other simply as verification in Supplement.

-Stating that if the study period were only longer the difference would have reached significance seems rather speculative - suggest softening this language (lines 294-297).

-Probably worth a more nuanced discussion of the disparate findings between overall ITS and the multivariate adjusted analysis. Conceptually, ITS does better at adjusting for pre-existing trends and I think this seems to be part of the lack of significant findings when looking at Figure 2 - since there was already a pre-policy trend towards increasing pediatric transfers. This is likely worth acknowledging as a caveat.

-Getting back to my earlier question about testing for difference in trends between adults and pediatrics, I do still find myself wondering whether the interaction test for difference in post-policy change trends would suggest a difference since the adult pattern leveled off but pediatrics did not. This result at least would be more consistent with the difference noted in the multivariate analysis.

-The authors may consider noting that because the multivariate analysis did not abrogate the post-policy association with pediatric transfers, they do at least have some evidence the increase over time is not solely due to an changing demographics/conditions.

9) On data availability: Given PLOS One's data availability clause, are at least select aggregate data available for upload? E.g., surely numerator/denominator of transfers each month by adult and pediatric patients could be made available as this has no direct tie to any identifiers?

Minor comments:

1) Abstract - results: would avoid use of "increase" or "decrease" unless statistically significant. Might instead prefer simply stating the number of transfers for each period.

2) Introduction - line 55: Was this meant to read "In 2024" rather than "In 2004"?

3) There are several areas where wording could be edited for clarity/brevity:

- Lines 64-66

- Line 71

4) Table 1:

-Defer to the Journal's specific statistical style recommendations, but I would usually refrain from reporting p values for baseline variable tables (to avoid false impression that these are hypothesis driven tests). These could either be simply omitted, or if authors feel it is important to display a measure of imbalance, standardized mean differences might provide the necessary information while avoiding the use of p values in a demographics table.

-Suggest reporting by each individual KTAS score rather than lumping 1-3 and 4-5 unless there is clear justification for this grouping

Reviewer #2: Thank you for conducting this research. I have a few quetions about gaps that were not addressed in the study.

You briefly mentioned that there was huge decrease between the number of patients from pre-policy to post-policy. Does the change in sample size affect the statistical tests? You briefly discussed limitations of the study in the discussion session. Secondly, you mentioned that the study was conducted in one pediatric emergency center. Where there other events or policy changes that might contributed to the reduction in sample size. Did you consider other factors such as secular trends or other historial events that might have contributed to increase in pediatric transfers or decrease in junior doctors?

6. PLOS authors have the option to publish the peer review history of their article (what does this mean?). If published, this will include your full peer review and any attached files.

Reviewer #1: **Yes:** Nicholas Turner

Reviewer #2: No

---

## [Author Response · Author response to Decision Letter 1]

4 Dec 2025

General Response

We sincerely thank the Academic Editor and both reviewers for their thoughtful and constructive comments.

We carefully revised the manuscript to address all major and minor points.

Key improvements include:

Additional diagnostics and model refinements in the interrupted time series (ITS) analysis, including Durbin–Watson and Breusch–Godfrey tests, heteroskedasticity- and autocorrelation-consistent (HAC) standard errors, and generalized least squares (GLS) with AR(1) structure.

Sensitivity analyses excluding an influential outlier month (December 2024).

Clear specification of separate adult and pediatric logistic regression models, with both unadjusted and adjusted ORs presented in Table 3.

Re-organization of the inverse-probability-of-treatment-weighting (IPTW) analysis as a supporting verification rather than a main analysis.

Expanded Discussion addressing the methodological distinction between logistic and ITS approaches and updated Limitations acknowledging sample-size effects and single-center design.

Preparation of a de-identified aggregate dataset compliant with PLOS ONE’s data-availability policy.

All tracked revisions have been accepted, and both a clean and tracked version are provided.

Journal Requirements:

The manuscript follows PLOS ONE style and file-naming conventions.

2. In the online submission form, you indicated that [The dataset analyzed in this study is not publicly available due to the policy of the institutional review board of Bundang CHA Medical Center. However, datasets are available from the corresponding author upon reasonable request.].

Aggregate de-identified dataset uploaded in accordance with journal policy.

“Ethics Statement” is now included in the Methods section.

1. Is the manuscript technically sound, and do the data support the conclusions?

Reviewer #1: Partly

Reviewer #2: Partly

2. Has the statistical analysis been performed appropriately and rigorously?

Reviewer #1: Yes

Reviewer #2: Yes

3. Have the authors made all data underlying the findings in their manuscript fully available?

Reviewer #1: No

Reviewer #2: Yes

4. Is the manuscript presented in an intelligible fashion and written in standard English?

Reviewer #1: No

Reviewer #2: Yes

5. Review Comments to the Author

Point-by-point responses to reviewers' comments:

Reviewer #1: The authors present 3 inter-related analyses of hospital transfers before and after a key policy change which led to an acute workforce shortage among South Korean physicians in February 2024. Interrupted time series did not find any significant change in pediatric transfers after the policy change (though there did appear to be a pre-existing upward trend); in contrast, multivariable logistic regression and IPTW did find that the post-policy time period seemed to be associated with an increase in transfer rate (allbeit no longer accounting for the pre-existing upward trend).

I think the work is overall an interesting opportunity to analyze the real world effects of a discrete policy change. It is both well suited to time series analysis and I think reasonable/appropriate to include at least one other analysis to adjust for potential changes in patient traits over time - though I think conducting both IPTW and multivariate regression is a bit duplicative as both adjust for this issue. The exploration of patterns of systemic disease categories pre and post policy change becomes quite speculative/exploratory, though at least p values were aggressively adjusted for (BH method given risk of false discovery).

I do think a number of areas could benefit from extra clarification/cleanup:

1) On selection of 3 different analytical methods: This work certainly seems best suited to interrupted time series analysis (which was one of the methods undertaken). I am a bit curious about the specific reasons for using both of the other methods. To me it would seem that multivariable logistic regression and IPTW effectively seek to accomplish similar goals (e.g., assessing likelihood of transfer by pre- or post-policy period while adjusting for potential confounders among patient-level characteristics).

-I think interrupted time series makes good sense as the primary analysis method

-I would next typically pick either multivariate logistic regression or IPTW (rather than both) as the second analysis - and frame it more clearly as an attempt to address/minimize potential confounding by differences in patient populations.

-If the authors really want to show the effect of potential confounders, then the multivariate regression approach might make the most sense as it more explicitly shows these effects; if the authors are most interested in simply looking at the overall trend in transfers (which seems to be their stated goal), then IPTW alone might make sense.

-If the authors really wanted to present results from both multivariate logistic regression and IPTW, I would still pick one or the other for the main manuscript and present the other as an alternative approach (perhaps in Supplemental materials) more as a sensitivity analysis showing that both come up with similar results. This would avoid making it seem as though the two are truly fully separate analyses when I don't see them as such (I suspect the IPTW weighting model actually closely resembles the logistic regression model).

-On that note, I was not clear what variables were utilized for IPTW - it would be worth specifying

Authors’ Response: Thank you for your comment. We agree and have clarified that ITS was used as the primary approach to assess temporal changes, whereas logistic regression evaluated associations after adjusting for patient-level covariates. IPTW was retained only as a supporting verification, now moved to Supporting Information. The Methods section was revised accordingly.

2) Appreciate the specific reference to assessment for auto-correlation and use of robust standard errors (lines 133-135), though the authors do not actually mention whether there was evidence of auto-correlation. If there were evidence of autocorrelation, there are usually additional modeling steps needed to account for that.

Authors’ Response: Thank you for this comment. We added explicit results of Durbin–Watson and Breusch–Godfrey tests. The latter suggested AR(1) correlation in the adult series; therefore, GLS models with AR(1) structure and HAC (Newey–West) standard errors were applied. These revisions appear in Methods and Results sections. Figure 2 and Table 4B were updated to reflect autocorrelation-adjusted estimates.

3) Line 137: Would soften/avoid language such as "To verify causal inference" as no adjustment can ever fully verify causal inference. Might suggest alternative language such as "To mitigate or address potential confounding by differences in patient populations pre- and post-policy announcement..."

Authors’ Response: Thank you for pointing out the overly strong phrasing and suggesting a clearer alternative for it, we revised to “ To mitigate potential confounding by differences in patient populations pre- and post-policy announcement, we additionally performed IPTW”.

4) Table 2: The presentation of Table 2 makes me begin to wonder how age grouping was handled (Adult vs Pediatric) - were two separate models constructed for each age group? Or was age group considered as a variable?

-If part of the point of the article were that adult and pediatric trends differed, one would usually consider something like an interaction term to statistically test whether adult/pediatric trends differed. This would usually provide justification for producing separate models.

-Regardless of how age group were handled, many statistically inclined readers might like to see separate columns displaying the univariate (unadjusted) and multivariate (adjusted) ORs as this would help the reader understand the effect of adjustment for suspected confounders.

Authors’ Response: We agree that clarification on separate models is needed. Separate multivariable logistic regressions were conducted for adult and pediatric groups.

Also, table 3 now includes both unadjusted and adjusted ORs (95 %CIs) for each variable, including detailed breakdowns by initial KTAS score (1–5).

The Methods section now specifies the modeling approach.

5) Table 3A: I would suggest only showing the adjusted p values as these are more exploratory/subgroup based analyses.

Authors’ Response: Thank you for the comment, unadjusted p values were removed; only Benjamini–Hochberg-adjusted p values now remain.

6) Table 4: similar to above, I am curious if two separate models (adult and pediatric) were set a priori or if there were specific interest in demonstrating their trends differed (in which case again an interaction test might help assess whether these are statistically significantly different from one another).

Authors’ Response: We greatly appreciate this insightful suggestion. We believe this specific review gave us a chance to widen our perspective in analyzing and proving the difference between the trends from different groups. A pooled logistic model with a period × age-group interaction term was added. The interaction was significant (OR 2.44, 95 % CI 1.56–3.85, p < 0.001), indicating a differential post-policy effect between adults and children. Results are described in the Methods and Results sections.

7) Figure 2: There is a single month near what appears to be "2025-12" with a near 1.2% transfer rate. As this point at least visually looks as though it could serve as an influential outlier, consider sensitivity analysis of trend with exclusion of this point.

Authors’ Response: Thank you for this comment. After the review, authors agree that sensitivity analysis would be more beneficial to understand the overall trend. We re-estimated ITS models excluding December 2024. The direction and significance of results were unchanged, confirming robustness. These are presented in Results and Figure 2.

8) In discussion of results, some of the above suggestions regarding methods might also apply:

-As expected, it seems IPTW and multivariate logistic regression came to the same conclusion that rate of pediatric transfers increased post policy change. This would seem to support that these are not really fully independent analyses (as in comment 1 above), but perhaps ought to have only one presented as a main analysis and the other simply as verification in Supplement.

-Stating that if the study period were only longer the difference would have reached significance seems rather speculative - suggest softening this language (lines 294-297).

-Probably worth a more nuanced discussion of the disparate findings between overall ITS and the multivariate adjusted analysis. Conceptually, ITS does better at adjusting for pre-existing trends and I think this seems to be part of the lack of significant findings when looking at Figure 2 - since there was already a pre-policy trend towards increasing pediatric transfers. This is likely worth acknowledging as a caveat.

-Getting back to my earlier question about testing for difference in trends between adults and pediatrics, I do still find myself wondering whether the interaction test for difference in post-policy change trends would suggest a difference since the adult pattern leveled off but pediatrics did not. This result at least would be more consistent with the difference noted in the multivariate analysis.

-The authors may consider noting that because the multivariate analysis did not abrogate the post-policy association with pediatric transfers, they do at least have some evidence the increase over time is not solely due to an changing demographics/conditions.

Authors’ Response: Thank you for this comment. The following revisions have been made to address these points:

ITS vs logistic regression nuance: Added paragraph explaining that ITS accounts for pre-existing trends and autocorrelation, which attenuated the apparent post-policy effect, yielding more conservative results.

Persistence after adjustment: Expanded to note that the post-policy increase in pediatric transfers remained significant after adjustment, suggesting that this pattern is not solely explained by demographic or clinical shifts.

In accordance with the reviewer’s suggestions, we made several revisions in the Discussion. First, language implying that a longer study period “might have led to statistical significance” has been softened to avoid speculative interpretation.

9) On data availability: Given PLOS One's data availability clause, are at least select aggregate data available for upload? E.g., surely numerator/denominator of transfers each month by adult and pediatric patients could be made available as this has no direct tie to any identifiers?

Authors’ Response: Thank you for pointing this out. A fully anonymized dataset containing monthly transfer counts and denominators for adult and pediatric groups has been uploaded as Supporting Information. No individual-level identifiers are included.

Minor comments:

1) Abstract - results: would avoid use of "increase" or "decrease" unless statistically significant. Might instead prefer simply stating the number of transfers for each period.

Authors’ Response: Thank you for this comment. We revised wording to describe counts rather than “increase/decrease” where statistical significance was absent.

2) Introduction - line 55: Was this meant to read "In 2024" rather than "In 2004"?

Authors’ Response: We corrected “2004” → “2024”.

3) There are several areas where wording could be edited for clarity/brevity:

-

---

## [Decision Letter · Decision Letter 1]

7 Jan 2026

PONE-D-25-47181R1Impact of the 2024 Korean medical workforce crisis on transfers in a pediatric emergency center: including comparative analyses with adultsPLOS One

Dear Dr. Paek,

Thank you for submitting your manuscript to PLOS ONE. After careful consideration, we feel that it has merit but does not fully meet PLOS ONE’s publication criteria as it currently stands. Therefore, we invite you to submit a revised version of the manuscript that addresses the points raised during the review process.

We look forward to receiving your revised manuscript.

Kind regards,

Sonu Bhaskar, MD PhD

Academic Editor

PLOS One

Journal Requirements:

Reviewers' comments:

Reviewer's Responses to Questions

**Comments to the Author**

1. If the authors have adequately addressed your comments raised in a previous round of review and you feel that this manuscript is now acceptable for publication, you may indicate that here to bypass the “Comments to the Author” section, enter your conflict of interest statement in the “Confidential to Editor” section, and submit your "Accept" recommendation.

Reviewer #1: All comments have been addressed

2. Is the manuscript technically sound, and do the data support the conclusions?

Reviewer #1: Yes

3. Has the statistical analysis been performed appropriately and rigorously? 

Reviewer #1: Yes

4. Have the authors made all data underlying the findings in their manuscript fully available?

Reviewer #1: Yes

5. Is the manuscript presented in an intelligible fashion and written in standard English?

Reviewer #1: Yes

6. Review Comments to the Author

Reviewer #1: I appreciate the extensive extra work the authors undertook in response to my questions. All of my questions have been fully answered.

Only one minor consideration remains: since the responses make clear that ITS was the primary analysis method, would consider modifying current abstract (and perhaps some of discussion) to lead with the ITS results & discuss the other regression approaches secondarily (which seems to match the analysis plan a bit more closely).

7. PLOS authors have the option to publish the peer review history of their article (what does this mean?). If published, this will include your full peer review and any attached files.

Reviewer #1: **Yes:** Nicholas A. Turner

---

## [Author Response · Author response to Decision Letter 2]

7 Jan 2026

General Response

We sincerely thank the Academic Editor and the reviewer for their careful re-evaluation of our revised manuscript and for acknowledging the additional analyses and clarifications provided in response to the previous round of review. We greatly appreciate the positive assessment that our additional work has adequately addressed all prior concerns, and we are grateful for the constructive and thoughtful feedback that helped improve the clarity and presentation of the manuscript.

Reviewers' comments:

Reviewer's Responses to Questions

Comments to the Author

1. If the authors have adequately addressed your comments raised in a previous round of review and you feel that this manuscript is now acceptable for publication, you may indicate that here to bypass the “Comments to the Author” section, enter your conflict of interest statement in the “Confidential to Editor” section, and submit your "Accept" recommendation.

Reviewer #1: All comments have been addressed

2. Is the manuscript technically sound, and do the data support the conclusions?

Reviewer #1: Yes

3. Has the statistical analysis been performed appropriately and rigorously?

Reviewer #1: Yes

4. Have the authors made all data underlying the findings in their manuscript fully available? The PLOS Data policy requires authors to make all data underlying the findings described in their manuscript fully available without restriction, with rare exception (please refer to the Data Availability Statement in the manuscript PDF file). The data should be provided as part of the manuscript or its supporting information, or deposited to a public repository. For example, in addition to summary statistics, the data points behind means, medians and variance measures should be available. If there are restrictions on publicly sharing data—e.g. participant privacy or use of data from a third party—those must be specified.

Reviewer #1: Yes

5. Is the manuscript presented in an intelligible fashion and written in standard English? PLOS ONE does not copyedit accepted manuscripts, so the language in submitted articles must be clear, correct, and unambiguous. Any typographical or grammatical errors should be corrected at revision, so please note any specific errors here.

Reviewer #1: Yes

6. Review Comments to the Author Please use the space provided to explain your answers to the questions above. You may also include additional comments for the author, including concerns about dual publication, research ethics, or publication ethics. (Please upload your review as an attachment if it exceeds 20,000 characters)

Reviewer #1: I appreciate the extensive extra work the authors undertook in response to my questions. All of my questions have been fully answered. Only one minor consideration remains: since the responses make clear that ITS was the primary analysis method, would consider modifying current abstract (and perhaps some of discussion) to lead with the ITS results & discuss the other regression approaches secondarily (which seems to match the analysis plan a bit more closely).

Thank you for your comment. In response to the reviewer’s suggestion, we revised the abstract to lead with the interrupted time series (ITS) results, with regression-based analyses presented subsequently. No changes were made to the underlying analyses or conclusions.

---

## [Decision Letter · Decision Letter 2]

10 Apr 2026

PONE-D-25-47181R2Impact of the 2024 Korean medical workforce crisis on transfers in a pediatric emergency center: including comparative analyses with adultsPLOS One

Dear Dr. Paek,

Thank you for submitting your manuscript to PLOS ONE. After careful consideration, we feel that it has merit but does not fully meet PLOS ONE’s publication criteria as it currently stands. Therefore, we invite you to submit a revised version of the manuscript that addresses the points raised during the review process.

We look forward to receiving your revised manuscript.

Kind regards,

Inge Roggen, M.D., Ph.D.

Academic Editor

PLOS One

Journal Requirements:

Reviewers' comments:

Reviewer's Responses to Questions

**Comments to the Author**

1. If the authors have adequately addressed your comments raised in a previous round of review and you feel that this manuscript is now acceptable for publication, you may indicate that here to bypass the “Comments to the Author” section, enter your conflict of interest statement in the “Confidential to Editor” section, and submit your "Accept" recommendation.

Reviewer #1: All comments have been addressed

Reviewer #3: (No Response)

2. Is the manuscript technically sound, and do the data support the conclusions?

Reviewer #1: Yes

Reviewer #3: Yes

3. Has the statistical analysis been performed appropriately and rigorously? 

Reviewer #1: Yes

Reviewer #3: Yes

4. Have the authors made all data underlying the findings in their manuscript fully available?

Reviewer #1: Yes

Reviewer #3: Yes

5. Is the manuscript presented in an intelligible fashion and written in standard English?

Reviewer #1: Yes

Reviewer #3: Yes

6. Review Comments to the Author

Reviewer #1: Thank you for updating the abstract - I believe it now better reflects the hard work done, and all of my comments have been answered.

Reviewer #3: Dear Authors,

Thank you for submitting your manuscript entitled “Impact of the 2024 South Korean Junior Doctor Workforce Departure on Interhospital Transfers from a Pediatric Emergency Center: A Comparative Analysis with Adult Patients.” This is a timely and policy-relevant study examining the impact of the medical school quota expansion announcement and subsequent junior doctor resignations on emergency care, with a useful pediatric–adult comparison.

The topic is important, and the analytical approach, interrupted time series supplemented by multivariable regression and IPTW, is appropriate. The manuscript is generally well structured and offers relevant insight into pediatric vulnerability. However, several points need clarification to improve rigor and interpretability before publication.

The pre- (March 2023 to January 2024) and post-policy (March 2024 to January 2025) periods are appropriate for seasonality control. Please specify the exact number of months included in each period and confirm data completeness through January 2025. For the ITS analysis, clarify whether counts or rates were modeled, how inverse-variance weighting was implemented, and how multiple ICD-10 diagnoses per patient were handled. The rationale for excluding R codes in subgroup analysis should also be stated more clearly.

There is a discrepancy between the non-significant ITS findings and the significant associations observed in the regression and IPTW analyses for pediatric transfers (adjusted OR approximately 2.49 and 2.21). While this is acknowledged, it should be more explicitly framed in terms of methodological differences. The increase in pediatric transfers (48 to 89) despite a roughly 42 percent decline in visits is a key result and should be presented more prominently across the abstract, results, and conclusion. Please also review the unadjusted ORs in Table 3, particularly for KTAS categories, for any inconsistencies.

The discussion appropriately highlights pediatric vulnerability, shifts in transfer reasons, and the added burden of specialist accompaniment. A brief expansion on plausible mechanisms, such as more risk-averse decision-making or limited pediatric readiness in general emergency departments, would strengthen interpretation while maintaining caution around causality. A short comparison with existing work on the 2024 crisis would also help contextualize the findings. The limitations section may be slightly condensed, and a brief note on broader policy implications for maintaining pediatric emergency readiness would add value.

Overall, this is a relevant and potentially impactful study. Addressing these points will improve clarity and strengthen the manuscript.

7. PLOS authors have the option to publish the peer review history of their article (what does this mean?). If published, this will include your full peer review and any attached files.

Reviewer #1: **Yes:** Nicholas Turner

Reviewer #3: No

---

## [Author Response · Author response to Decision Letter 3]

10 Apr 2026

General Response

We sincerely thank the Academic Editor and the reviewer for their careful re-evaluation of our revised manuscript.

We greatly appreciate the positive assessment that our additional work has adequately addressed all prior concerns, and we are grateful for the constructive and thoughtful feedback that helped improve the clarity and presentation of the manuscript.

Reviewers' comments:

1. Reviewer #1: Thank you for updating the abstract - I believe it now better reflects the hard work done, and all of my comments have been answered.

We greatly appreciate the positive assessment.

Reviewer #3: Dear Authors,

Thank you for submitting your manuscript entitled “Impact of the 2024 South Korean Junior Doctor Workforce Departure on Interhospital Transfers from a Pediatric Emergency Center: A Comparative Analysis with Adult Patients.” This is a timely and policy-relevant study examining the impact of the medical school quota expansion announcement and subsequent junior doctor resignations on emergency care, with a useful pediatric–adult comparison.

The topic is important, and the analytical approach, interrupted time series supplemented by multivariable regression and IPTW, is appropriate. The manuscript is generally well structured and offers relevant insight into pediatric vulnerability. However, several points need clarification to improve rigor and interpretability before publication.

1. The pre- (March 2023 to January 2024) and post-policy (March 2024 to January 2025) periods are appropriate for seasonality control. Please specify the exact number of months included in each period and confirm data completeness through January 2025.

Thank you for this comment. We have added a sentence to the Methods section explicitly confirming the study period duration and data completeness: “Each study period therefore comprised 11 months (March through January), and data collection was confirmed to be complete through January 2025.”

2. For the ITS analysis, clarify whether counts or rates were modeled, how inverse-variance weighting was implemented, and how multiple ICD-10 diagnoses per patient were handled.

Thank you for this comment. We have revised the Statistical Analysis section to address these clarifications. First, the phrase “monthly counts of ED visits and transfers” was corrected to “monthly transfer rates (modeled as the proportion of transfers among total ED visits, using a quasibinomial generalized linear model).” Second, inverse-variance weighting (1/n_visits) was already described in the context of the GLS sensitivity analysis; no additional change was required. Third, we added an explicit description of how multiple ICD-10 diagnoses were handled.

3. The rationale for excluding R codes in subgroup analysis should also be stated more clearly.

Thank you for this comment. We have added explicit rationale for the exclusion of R codes to the Methods section.

4. There is a discrepancy between the non-significant ITS findings and the significant associations observed in the regression and IPTW analyses for pediatric transfers (adjusted OR approximately 2.49 and 2.21). While this is acknowledged, it should be more explicitly framed in terms of methodological differences.

Thank you for this important comment. We have substantially revised the corresponding paragraph in the Discussion to provide an explicit methodological explanation. Specifically, we clarified that ITS operates at the population level, modeling monthly aggregated transfer rates with only 11 data points per group, limiting statistical power for detecting modest changes; additionally, the substantial decline in pediatric ED visit volume (approximately 42%) introduces a denominator effect that the ITS model partially absorbs. In contrast, multivariable logistic regression and IPTW operate at the patient level and directly isolate the association between the period indicator and individual transfer probability after adjusting for case-mix confounders. The directional consistency across all three analytical approaches was highlighted as supporting the overall interpretation that the post-policy period was associated with a genuine increase in pediatric transfer risk.

5. The increase in pediatric transfers (48 to 89) despite a roughly 42 percent decline in visits is a key result and should be presented more prominently across the abstract, results, and conclusion.

Thank you for this comment. We agree that this contrast is a key finding of our study. We have added explicit reporting of the absolute numbers and visit volume decline in three locations.

6. Please also review the unadjusted ORs in Table 3, particularly for KTAS categories, for any inconsistencies.

Thank you for this comment. We carefully reviewed all unadjusted and adjusted ORs for KTAS categories in Table 3 and confirmed they are internally consistent and monotonically decreasing across KTAS levels for both adult and pediatric groups, with KTAS 1 (most severe) serving as the reference category and higher KTAS levels (lower acuity) showing progressively lower odds of transfer. No numerical inconsistencies were identified. To improve clarity in the Results text, we revised the phrase “less severe or acute” to “with KTAS 1 as the most severe reference category, such that lower acuity levels showed substantially reduced odds of transfer.”

7. The discussion appropriately highlights pediatric vulnerability, shifts in transfer reasons, and the added burden of specialist accompaniment. A brief expansion on plausible mechanisms, such as more risk-averse decision-making or limited pediatric readiness in general emergency departments, would strengthen interpretation while maintaining caution around causality. A short comparison with existing work on the 2024 crisis would also help contextualize the findings.

Thank you for this constructive suggestion. We have expanded the Discussion. First, we elaborated on plausible mechanisms by referencing existing literature on heightened risk-averse decision-making during workforce shortages, the limited readiness of general emergency departments to provide specialized pediatric care, and the inherently higher complexity of pediatric diagnosis and management. Second, we added a comparison with a recently published nationwide interrupted time series study, which similarly documented a sharp, sustained decline in ED visits following the junior physicians’ walkout.

8. The limitations section may be slightly condensed, and a brief note on broader policy implications for maintaining pediatric emergency readiness would add value.

Thank you for these comments. Regarding the limitations we condensed it with more concise sentences to reduce redundancy. Regarding policy implications: we added a dedicated paragraph to the Discussion presenting the broader policy implications of our findings. This paragraph emphasizes that abrupt and large-scale workforce policy changes without adequate transition planning risk creating disproportionate disruptions in vulnerable subgroups such as pediatric patients, and calls for minimum staffing thresholds, pre-defined transfer protocols, and the designation of pediatric emergency capacity as a protected resource during workforce crises.

Overall, this is a relevant and potentially impactful study. Addressing these points will improve clarity and strengthen the manuscript.

Thank you for your comment. In response to the reviewer’s suggestion, we revised the manuscript.

---

## [Editor Report · Decision Letter 3]

13 Apr 2026

Impact of the 2024 Korean medical workforce crisis on transfers in a pediatric emergency center: including comparative analyses with adults

PONE-D-25-47181R3

Dear Dr. Paek,

We’re pleased to inform you that your manuscript has been judged scientifically suitable for publication and will be formally accepted for publication once it meets all outstanding technical requirements.

Kind regards,

Inge Roggen, M.D., Ph.D.

Academic Editor

PLOS One
---

## [Editor Report · Acceptance letter]

PONE-D-25-47181R3

PLOS One

Dear Dr. Paek,

I'm pleased to inform you that your manuscript has been deemed suitable for publication in PLOS One. Congratulations! Your manuscript is now being handed over to our production team.

Kind regards,

on behalf of

Prof. Inge Roggen

Academic Editor

PLOS One